# Information needs among women taking part in primary HPV screening in England: a content analysis

Laura Marlow [iD] ,[1,2] Alice S Forster [iD] ,[2] Emily McBride,[2] Lauren Rockliffe,[3] Henry Kitchener,[4] Jo Waller [iD] [1,2]

[1]Cancer Prevention Group, School of Cancer and Pharmaceutical Sciences, King's College London, London, UK
[2]Research Department of Behavioural Science and Health, University College London, London, UK
[3]School of Health Sciences, Faculty of Biology, Medicine and Health, The University of Manchester, Manchester, UK
[4]Women's Cancer Centre, Institute of Cancer Sciences, University of Manchester, Manchester, UK

**Correspondence to**
Dr Laura Marlow;
l.marlow@kcl.ac.uk

## ABSTRACT

**Objectives** Introducing primary human papillomavirus (HPV) testing to cervical screening programmes means changes to the results women receive. We explored additional information needs among women undergoing HPV primary screening.

**Design** Women were sent a postal questionnaire shortly after receiving their results and 6 and 12 months later. Each questionnaire asked if women had any unanswered questions about cervical screening or HPV testing. Free-text responses constituted the data. Themes were identified using content analysis.

**Setting** National Health Service (NHS) Cervical Screening Programme, England.

**Participants** 381 women who recorded one or more free-text responses.

**Results** The most common theme represented women's emotional responses and attempts to understand their results. This theme was raised by 45% of women overall, but was as high as 59% in the HPV cleared group. General questions about the cause and epidemiology of HPV were raised by 38% of women and were more common among those testing HPV positive with normal cytology (52%). Questions about the purpose and procedure for HPV testing were most common among HPV-negative women (40%, compared with 16%–24% of the other results groups). Questions about future implications of test results were raised by 19% of women, and this theme was most common among those with persistent HPV.

**Conclusions** Despite provision of information alongside screening invitations, women can still have unanswered questions following receipt of their results. Details about the epidemiology of HPV and why cervical screening procedures are changing should be included with screening invitations. Some results groups may benefit from additional tailored information with their results letter.

## INTRODUCTION

Cervical screening programmes have traditionally involved looking for abnormal cytology but human papillomavirus (HPV) primary screening can provide many benefits[1] and has already replaced cytology-based cervical screening in England, Australia, the Netherlands and Wales, and several other countries are expected to follow in the coming years.[2 3] HPV testing looks for presence of the

**Strengths and limitations of this study**

► The study benefits from the inclusion of participants who had been tested for human papillomavirus (HPV) as part of routine HPV primary cervical screening.
► Participants had a range of HPV and cytology screening results which allowed us to compare information needs between results groups.
► Two-thirds of women did not leave a free-text response.
► Questionnaires were completed at least 2 weeks after receiving results so participants may already have sought additional information.
► Those with less education were less likely to leave a free-text comment.

HPV virus. Where HPV is found the sample is looked at for cytology. Women with HPV positive/abnormal cytology results are referred for colposcopy, women with HPV positive/normal cytology are recalled 12 months later. Based on a large pilot study in England, around 13% of women aged 25–64 years will be told they are HPV positive,[4 5] compared with ~6% who currently receive an abnormal cytology result (with or without HPV). These figures are expected to decrease somewhat as cohorts offered HPV vaccination move into the programme.[6] Nevertheless, many women will be receiving an HPV-positive result, warranting careful consideration of how these results are communicated.

Of particular concern is that some women will be learning about the link between cervical cancer and a sexually transmitted infection for the first time, which may come as a shock and could raise concerns about sexual relationships.[7–9] In a review of studies exploring understanding of HPV and information needs,[10] women found it difficult to incorporate new information about HPV testing into their pre-existing understanding of cervical screening and often sought additional information after being told they were

HPV positive. The review identified uncertainty about HPV transmission, prevention, symptoms, risk factors (for HPV and cervical cancer), whether HPV could cause other cancers, treatment, fertility and the natural history of the virus. However, most of the studies included were small qualitative studies carried out before 2007, half of which used samples of women who had not actually been tested for HPV. More recently, qualitative interviews with women who were told they were HPV positive in an HPV self-sampling trial identified some key themes: intense affect (feelings and emotions) after receiving positive results, importance of discussing results with a provider, information seeking, confusion about purpose and meaning of HPV versus Pap tests.[11]

Establishing women's information needs in the context of primary screening is vital to inform patient education and communication strategies. Clear information provided at the appropriate time point (eg, alongside results) may help to minimise the adverse psychological responses to HPV-positive results that have been identified.[12] The aims of this study were to (1) identify the information needs of women participating in primary HPV screening and (2) explore how these might vary according to women's HPV and cytology results. This study is part of a broader psychological evaluation of HPV primary testing in England.[13]

## METHODS
### Participants
Participants were women aged 24–65 years who attended for cervical screening in England in one of five sites piloting HPV primary testing (between 2016 and 2017). Women testing HPV negative were invited for routine recall, whereas those testing positive had reflexive cytology and were managed accordingly (see online supplemental file 1 for a flow diagram and additional contextual information about cervical screening in England).

Recruitment was stratified to ensure data were collected from women receiving different screening results (see online supplemental figure 2): (1) negative for HPV, (2) HPV positive with normal cytology and (3) HPV positive with abnormal cytology. We also recruited two groups of women who had initially tested positive for HPV with normal cytology, and who had reattended at 12-month follow-up and either (4) had persistent HPV that is, they were still HPV positive with normal cytology or (5) had cleared the infection that is, they now tested HPV negative. A group of women undergoing conventional cytology screening were also recruited but we have excluded their data from the present analyses.

### Procedure
The data reported here were collected from cross-sectional surveys sent to women at three time points: shortly after receiving their results (baseline), 6 months and 12 months. The full protocol is available elsewhere,[13] but in brief women were contacted by post within 2 weeks

of receiving their screening results letter, and invited to complete and return a consent form and questionnaire. Women who returned the questionnaire were also sent questionnaires 6 and 12 months later. The primary outcome measures assessed in the questionnaire were anxiety and general distress.[12]

### Patient and public involvement
Neither patients nor the public were involved in the design, conduct, analysis or interpretation of this study.

### Measures
At each of the three time points, women were asked 'Do you have any unanswered questions about cervical screening or HPV testing?' and space for an open response was provided. Free-text responses to this question constituted the primary data for analyses. At the end of the baseline questionnaire there was also a space provided for 'any other comments'. Free-text responses recorded here were also included where they were relevant to the aims of this analysis (irrelevant comments were excluded, for example comments about practical aspects of the survey study). Sociodemographic information including age, marital status, education and ethnicity were also collected. Information on women's screening results was collected directly from the screening laboratories.

### Analysis
Content analysis was used to explore women's free-text responses. Responses from all three time points were analysed together. Content analysis can be defined as 'subjective interpretation of the content of text data through the systematic classification process of coding and identifying themes'.[14] Responses were typed into an excel spreadsheet and two authors (LR and EM) immersed themselves in the baseline data. An initial coding frame was developed using an inductive, conventional content analysis approach (ie, avoiding preconceived categories).[15] Three senior members of the research team (LM, AF, JW) then coded the data for 20% of the baseline participants (n=60), before refining the coding frame. All responses (from each time point) were then independently double coded. Any discrepancies were discussed. Multiple codes were allocated to individual responses when appropriate. We used 2-by-5 $X^2$ tests to explore differences in the proportion of women citing each major theme by result group. Subthemes are reported descriptively.

## RESULTS
Overall 921 women who had undergone HPV primary screening returned their baseline questionnaire (online supplemental figure 1). A total of 507 free-text responses were recorded (baseline=329/921, 6 months=110/762 and 12 months=68/537). Women testing HPV positive with normal cytology and those with persistent HPV were most likely to leave a free-text response (50%); HPV-negative women were least likely (26%) to do so. Free-text

 Marlow L, et al. BMJ Open 2020;**10**:e044630. doi:10.1136/bmjopen-2020-044630

**Table 1** Sample characteristics of women participating in HPV primary testing who did and did not record a free-text response during the course of the study (n=921)

| | Free-text response recorded at any time | | No free-text response recorded | | |
|---|---|---|---|---|---|
| | N | Row % | n | Row % | χ²(df), p value |
| Overall | 381 | 41.4 | 540 | 58.6 | |
| Result group | | | | | |
| HPV negative | 65 | 26.2 | 183 | 73.8 | χ²(4)=38.49, <0.001 |
| HPV positive, cytology normal | 126 | 50.0 | 129 | 50.0 | |
| HPV positive, cytology abnormal | 67 | 39.4 | 103 | 60.6 | |
| Persistent HPV | 91 | 50.8 | 88 | 49.2 | |
| Cleared HPV | 29 | 43.9 | 37 | 56.1 | |
| Age (years) | | | | | |
| 24–34 | 171 | 46.7 | 195 | 53.3 | χ²(3)=10.63, 0.014 |
| 35–44 | 66 | 32.7 | 136 | 67.3 | |
| 45–54 | 80 | 41.0 | 115 | 59.0 | |
| 55–65 | 64 | 41.0 | 92 | 59.0 | |
| Marital status* | | | | | |
| Current partner | 281 | 40.7 | 410 | 59.3 | χ²(1)=0.79, 0.375 |
| No partner | 95 | 44.4 | 119 | 55.6 | |
| Education† | | | | | |
| Degree or higher | 189 | 48.8 | 198 | 51.2 | χ²(2)=14.62, 0.001 |
| Qualifications below degree | 177 | 36.0 | 314 | 64.0 | |
| No formal qualifications | 7 | 38.9 | 11 | 61.1 | |
| Ethnicity | | | | | |
| White (British or other) | 356 | 42.7 | 527 | 57.3 | χ²(1)=4.77, 0.029 |
| Other ethnicity‡ | 20 | 28.6 | 50 | 71.4 | |

Where n does not add up to n=921, this is due to missing data.
*Marital status: current partner (married, civil partnership, living with partner, in a relationship) and no partner (single, divorced, widowed).
†No formal qualifications included those with no qualifications and those who were still studying with no previous qualifications.
‡Other ethnicity includes: Asian/Asian British, Black/African/Caribbean/Black British, mixed/multiple ethnic groups, other ethnic group.
HPV, human papillomavirus.

responses were also more common among women in the youngest age group and those with a degree qualification (see table 1 for sample characteristics). We have described each theme below with the prevalence of themes reported in table 2 (overall and by results group). Illustrative examples of women's comments are presented in table 3.

### Reaction to and understanding of results

Across all results groups (except HPV negative), the most common theme was 'Reaction to and understanding of results'. This theme was most frequently present in comments made by HPV-positive women (with normal or abnormal cytology, 51% and 54%, respectively) and by women who had cleared HPV at 12 months (59%). Women expressed a wide range of emotional responses to their results including shock, worry and relief. Comments included questions about the exact meaning of their result, including clarification about which HPV type they had. Implications for sexual relationships were raised by a number of women, including requests for clarification of what their result meant for future sexual relationships,

concern about reinfection within a relationship and the possible consequences of infection for their partner. A lack of confidence in HPV results and requests for cytology were recorded by 5% of women (12% of women who were HPV negative and 21% had cleared HPV). Having previously experienced an abnormal result and approaching the end of cervical screening eligibility were reasons that women gave for concern about not having a cytology test.

### Questions about HPV and cervical cancer

Over one-third of women who left a comment recorded a question about HPV (38%) and this was more common among women who were HPV positive with normal cytology (52%) or who had persistent HPV (41%). Women asked about various aspects of HPV epidemiology including questions about the timeline of infection, latency and clearance. Women's questions about HPV also included requests for clarification about the cause of their HPV, frequently including references to their long-term or sexual relationships. The potential for preventing

**Table 2** Number of women mentioning each major and subtheme overall and by test result group

| | Overall | HPV negative | HPV positive, cytology normal | HPV positive, cytology abnormal | Persistent HPV | Cleared HPV | $\chi^2$(df), p value |
|---|---|---|---|---|---|---|---|
| | (n=381) | (n=65) | (n=129) | (n=67) | (n=91) | (n=29) | |
| | n (%) | n (%) | n (%) | n (%) | n (%) | n (%) | |
| **Reaction to and understanding of results** | **170 (45)** | **11 (17)** | **69 (54)** | **34 (51)** | **39 (43)** | **17 (59)** | $\chi^2$(4)=27.72, 0.001 |
| Emotional response | 85 (22) | 5 (8) | 28 (22) | 24 (36) | 18 (20) | 10 (35) | |
| Meaning of results | 72 (19) | 1 (2) | 35 (27) | 11 (16) | 21 (23) | 4 (14) | |
| Impact on sexual relationships | 41 (11) | 0 | 25 (19) | 4 (6) | 11 (12) | 1 (3) | |
| Confidence in results | 20 (5) | 8 (12) | 4 (3) | 1 (2) | 1 (1) | 6 (21) | |
| **Questions about HPV** | **143 (38)** | **8 (12)** | **67 (52)** | **20 (30)** | **37 (41)** | **11 (38)** | $\chi^2$(4)=31.13, <0.001 |
| General lack of understanding | 24 (7) | 2 (3) | 15 (12) | 6 (9) | 4 (4) | 0 | |
| Epidemiology of HPV | 63 (17) | 2 (3) | 30 (23) | 5 (8) | 20 (22) | 6 (21) | |
| Cause of HPV | 32 (8) | 1 (2) | 20 (16) | 3 (5) | 6 (7) | 2 (7) | |
| Prevention/treatment of HPV | 28 (7) | 0 | 11 (9) | 7 (10) | 9 (10) | 1 (3) | |
| HPV vaccination | 39 (10) | 3 (5) | 20 (16) | 2 (3) | 11 (12) | 3 (10) | |
| **Questions about cervical cancer** | **52 (14)** | **5 (8)** | **24 (19)** | **7 (10)** | **14 (15)** | **2 (7)** | $\chi^2$(4)=6.58, 0.160 |
| Risk of cervical cancer | 45 (12) | 3 (3) | 20 (16) | 7 (10) | 14 (15) | 2 (7) | |
| Other cervical cancer risk factors | 14 (4) | 3 (5) | 7 (5) | 1 (2) | 2 (2) | 1 (3) | |
| **Purpose and procedure for HPV testing** | **85 (22)** | **26 (40)** | **20 (16)** | **16 (24)** | **16 (18)** | **7 (24)** | $\chi^2$(4)=16.51, 0.002 |
| Purpose | 22 (6) | 11 (17) | 5 (4) | 4 (6) | 2 (2) | 0 | |
| Procedure | 18 (5) | 10 (15) | 3 (2) | 1 (2) | 0 | 4 (14) | |
| Timing | 36 (9) | 4 (6) | 12 (9) | 5 (8) | 11 (12) | 4 (14) | |
| Delivery of results | 15 (4) | 4 (6) | 1 (1) | 7 (10) | 3 (3) | 0 | |
| **Future implications of test results** | **73 (19)** | **1 (2)** | **30 (23)** | **10 (15)** | **27 (30)** | **5 (17)** | $\chi^2$(4)=21.76, <0.001 |
| Clinical management | 39 (10) | 1 (2) | 14 (11) | 4 (6) | 19 (21) | 1 (4) | |
| Fertility/sexual health | 13 (3) | 0 | 5 (4) | 1 (3) | 3 (3) | 3 (10) | |
| Advice on clearing HPV | 18 (5) | 0 | 10 (8) | 2 (3) | 5 (6) | 1 (3) | |
| Testing for partners | 7 (2) | 0 | 4 (3) | 2 (3) | 1 (1) | 0 | |
| **Information seeking/(di)satisfaction** | **85 (22)** | **8 (12)** | **37 (29)** | **14 (21)** | **22 (24)** | **4 (14)** | $\chi^2$(4)=8.25, 0.083 |
| Information seeking | 30 (8) | 3 (5) | 15 (12) | 7 (10) | 5 (6) | 0 | |
| (Di)satisfaction | 73 (19) | 6 (9) | 31 (24) | 12 (18) | 20 (22) | 4 (14) | |

HPV, human papillomavirus.

future HPV infections and treating current ones was also raised. Some women expressed a more general lack of understanding about HPV, saying they had never heard of it or did not know what it was.

A smaller number of women provided comments about cervical cancer (14% across all results groups). A range of general questions were raised about the risk of developing cervical cancer. Some women also asked about other specific causal risk factors for cervical cancer (eg, polycystic ovary syndrome, contraceptive implants, previous cancer diagnosis or treatment).

### Purpose and procedure for HPV testing

'Purpose and procedure for HPV testing' was the most common theme for HPV-negative women (40%) but was also raised by HPV-positive women (16%–24%) and women who had cleared HPV (24%). Questions about the purpose of HPV testing were predominantly to clarify how HPV testing fit with their existing knowledge of cervical screening, but some women mentioned being unaware they had been tested for HPV until they received their results. Some HPV-negative women wanted to clarify

**Table 3** Examples of each quote

| | |
|---|---|
| **Reaction to and understanding of results** | |
| Emotional response | "I am very worried in case I end up with cervical cancer" (HPV+, cyto norm; 55–65 years)<br>"I feel quite distressed about the results and the letter … has caused me stress and anxiety" (HPV+, cyto abnorm; 24–34 years)<br>"I was advised I do not have HPV, I have had this persistently for years, I am so relieved" (HPV cleared; 24–34 years) |
| Confidence in results | "I have a family history of abnormal cells being found, but I was not tested for anything other than HPV. I would like to have a further test to confirm no abnormal cells" (HPV−; 35–44 years)<br>"Because I have previously had abnormal cells… I was not reassured by my HPV-negative result" (HPV−; 45–54 years)<br>"I am uneasy about the fact that cells have not been checked for abnormality, especially as no further tests will be offered to me" (HPV−; 55–65 years) |
| Meaning of results | "I was cleared of HPV last year, why has it come back?" (HPV+, cyto norm; 45–54 years)<br>"I have had two smears now both HPV-positive. How long can a person be HPV-positive for HPV?" (HPV persistent; 45–54 years)<br>"My previous test was positive and this one was negative. Does this mean it is still present but not active?" (HPV cleared; 24–34 years)<br>"I caught genital warts at 23 - is this somehow different?" (HPV+, cyto norm; 24–34 years) |
| Impact on sexual relationships | "Not sure what this means for future sexual relationships" (HPV+, cyto abnorm; 24–34 years)<br>"Can it be perpetuated by continuously being passed from one partner to the other?" (HPV+, cyto norm; 55–65 years)<br>"Should I tell sexual partners?" (HPV+, cyto norm; 24–34 years)<br>"Initially I worried about what my husband would think" (HPV persistent; 55–65 years)<br>"I blame my partner for this" (HPV+, cyto norm; 35–44 years) |
| **Questions about HPV** | |
| General lack of understanding | "I didn't even know I had been tested for HPV. Have never heard of it before" (HPV+, cyto norm; 35–44 years)<br>"I don't really understand what HPV is" (HPV+, cyto norm; 24–34 years) |
| Epidemiology of HPV | "How long does HPV last? What will happen if it doesn't go away?" (HPV+, cyto norm; 24–34 years)<br>"Has it gone and come back again or have I had it for 3 years?" (HPV+, cyto norm; 35–44 years)<br>"I'm still unclear as to what makes some peoples CIN1 cells disappear while others develop further" (HPV+, cyto abnorm; 35–44 years)<br>"Will it ever go away? Or get worse?" (HPV persistent; 55–65 years) |
| Cause of HPV | "I don't understand how I have got HPV" (HPV+, cyto norm; 24–34 years)<br>"I have not been sexually active for 6 years and can't understand why I have got it with only having one long-term partner" (HPV+, cyto abnorm; 35–44 years) |
| Prevention/treatment for HPV | "Should I now always use condoms for sex?" (HPV+, cyto norm; 24–34 years)<br>"Is there something I can take to get rid of HPV?" (HPV+, cyto norm; 35–44 years)<br>"Are there really no ways to treat the HPV virus?" (HPV persistent; 55–65 years) |
| HPV vaccination | "I had the HPV vaccine, why didn't it work?" (HPV+, cyto norm; 24–34 years)<br>"I'd like to know if I could be offered the vaccine and whether it would work for me" (HPV+, cyto norm; 35–44 years)<br>"I have been considering having the vaccine but unsure of benefits at my age" (HPV cleared; 35–44 years) |
| **Questions about cervical cancer** | |
| Risk of cervical cancer | "What are the chances of this becoming cancerous?" (HPV+, cyto norm; 55–65 years)<br>"What proportion of women who have had 2 smears detecting high risk HPV will go on to develop cervical cancer?" (HPV persistent; 45–54 years) |
| Other cervical cancer risk factors | "There is a vast history of cancer in my family. Am I more likely to get cancer?" (HPV+, cyto norm; 55–65 years)<br>"I have a contraception implant - does this affect HPV or my chances of developing cervical cancer" (HPV+, cyto norm; 24–34 years) |
| **Purpose and procedure for HPV testing** | |

**Table 3** Continued

| | |
|---|---|
| Purpose | "I am not sure if HPV test covers more, less or the same as a normal smear test" (HPV−; 45–54 years)<br>"I do not know if one is more thorough and effective than the other" (HPV−; 35–44 years)<br>"Was given no information that would be a different test other than smear" (HPV+, cyto norm; 45–54 years) |
| Procedure | "Was the HPV an additional test in addition to a normal smear test?" (HPV−; 45–54 years)<br>"Why when HPV is not present they don't test the sample" (HPV cleared; 24–34 years) |
| Timing | "Why can't I be re-tested in 6 months instead of waiting another year?" (HPV+, cyto norm; 45–54 years)<br>"Is having my next smear in 1 year soon enough? Could my cells change quickly enough to be cancerous before then?" (HPV+, cyto norm; 45–54 years)<br>"I would like to be reassured that the intervals between tests are adequate to pick up any changes to my body" (HPV cleared; 35–44 years) |
| Delivery of results | "I haven't received a letter with my results and I don't ever recall receiving results" (HPV−; 35–44 years)<br>"I had lots of questions that I could not get answered because results come in letter form" (HPV+, cyto abnorm; 24–34 years) |
| Future implications of test results | |
| Clinical management | "My test was positive two times and I want to meet a specialist" (HPV+, cyto norm; 45–54 years)<br>"I had a second positive HPV and have not been invited for further testing which the nurse said I would be. I am wondering why" (HPV persistent; 55–65 years)<br>"Am I now having a colposcopy because I have had HPV for 2 years?" (HPV persistent; 45–54 years) |
| Fertility/sexual health | "I am due a second test in 1 years time, but I am hopefully aiming to be pregnant around then, is this a major problem?" (HPV+, cyto norm; 24–34 years)<br>"Will it increase my chances of miscarriage?" (HPV persistent; 24–34 years) |
| Advice on clearing HPV | "What can I do in the next 12 months to help myself?" (HPV+, cyto norm; 24–34 years)<br>"Is there anything I can do to stop HPV developing into cancer?" (HPV persistent; 24–34 years) |
| Testing for partners | Why men don't get test for it if they can transmit it? (HPV+, cyto abnorm; 24–34 years) |
| Information seeking and (di)satisfaction | |
| Information seeking | "I contacted my GP for more information" (HPV+, cyto norm; 24–34 years) |
| (Di)satisfaction | "It's not explained in a very useful manner" (HPV persistent; 24–34 years)<br>"I want to have more information about HPV" (HPV+, cyto norm; 45–54 years)<br>"On receiving letter about results I felt I had lots of questions that I could not get answered" (HPV+, cyto abnorm; 24–34 years) |

GP, general practitioner; HPV, human papillomavirus.

whether their sample had been cytology tested or why cytology was not also performed.

Women also made comments about the timing of tests, particularly in relation to repeat HPV testing. They wanted to be reassured that the recommended intervals were 'soon enough' or 'adequate'. Several women also commented on the delivery of their results, for example saying that the results had not been clear from the letter they received and that further discussion with their general practitioner (GP) had been needed.

### Future implications of test results

The theme 'Future implications of the results' was identified in 19% of the comments and was most commonly recorded for women with persistent HPV (30%). These comments related specifically to clinical management, with requests for clarification about what would happen next for them. Implications for fertility, for their partners being tested and advice on clearing HPV were mentioned by a few women.

### Information seeking and (di)satisfaction

Some women described their experiences of seeking additional information about HPV. This predominantly included experiences of searching online or contacting their GP surgery to discuss their result further (with a GP or nurse) and was recorded for 22% of those leaving a comment. A number of women indicated satisfaction, or more commonly lack of satisfaction, with the information they had received.

### DISCUSSION

This study found women undergoing primary HPV testing for cervical screening can have additional information needs after receiving their results. Requests for more information about the epidemiology and cause of HPV were common across all results groups, so this seems to be important information to communicate to women taking part in screening. Other questions were more common among women receiving particular results; for example women receiving an HPV positive result (with normal cytology) often had questions about the meaning of this result and wanted advice

about the implications for sexual relationships. Women with abnormal cytology seemed to have fewer questions about HPV and the meaning of their result. This may in part be because they had been referred for colposcopy and even if they had not yet attended and had the opportunity to ask questions, they would have received an additional information leaflet with their results letter. They did, however, express more worry and concern. These differences suggest there may be merit in including results-tailored information alongside the delivery of results. However, the wide range of themes identified and the personalised nature of many questions mean signposting to additional information will also be important.

Some of our themes relating to women's understanding of HPV and cervical cancer were similar to other studies,[9 10] supporting the need to provide women with information about the cause and epidemiology of HPV. Women's desire to develop a coherent model of what HPV is, the timeline of infection and its cause and consequences is supported by theoretical models of illness representation which suggest that these aspects are important for understanding HPV and cervical cancer, and consequently for coping with being given an HPV-positive result.[16 17]

Previous studies have suggested that women are often shocked to learn about the sexually transmitted nature of HPV.[7 8] 'Implications for sexual relationships' was not the most common theme identified and while this question was raised by some women (particularly those who test HPV positive with normal cytology), it is reassuring that this was not more widespread in women's responses. Some themes such as the impact of HPV on fertility or questions about the impact for male partners, were raised by very few women suggesting these are unlikely to be major areas of concern. Studies exploring the psychological impact of testing HPV positive in the context of organised screening show no differences in distress across results groups, but anxiety can be slightly higher in HPV-positive women, at least in the short term.[12 18]

The main strength of this study is that we included women who had been tested for HPV as part of routine cervical screening, meaning we were able to compare responses across results groups. However, there are some limitations. The overall response rate for the questionnaire was low and of those responding, less than half recorded a free-text response. We cannot be sure if women who chose not to leave a free-text response had no information needs or just did not state them. In addition, since the questionnaires were completed at least 2 weeks after receiving results, women may already have sought answers to any questions they initially had. It is therefore likely that our study underestimates the number of questions women have on receiving their result. Women from lower socioeconomic backgrounds were less likely to return the questionnaire[12] and those with less education were less likely to leave a free-text comment. It is therefore possible that the results under-represent the concerns of women from lower socioeconomic backgrounds. Future research on HPV information needs should focus on these harder to reach groups.

A number of implications arise from this study. First, it is important that women are made aware of HPV before being tested. For some women, including information about HPV in invitation letters will not be sufficient so sample-takers have an important role in ensuring women know they are having an HPV test. In some instances, this may be simply by drawing their attention to the information leaflet that they receive with their screening invitation, but for some women this will lead to additional questions which sample-takers should be prepared to answer.

Second, information provided to women alongside their results should ideally be tailored to the result being communicated. Many of the women who were HPV positive or had recently cleared HPV had questions about the meaning of their result and some described contacting their GP surgery to discuss this. This is consistent with the findings in the USA, where women receiving HPV-positive results felt a sense of urgency to discuss it with their healthcare provider and felt reassured after this had happened.[11] It is important that staff in primary care are well equipped to answer women's questions or to direct them to the best source of information. This may involve answering questions about HPV themselves or directing women to online information materials (eg, the National Health Service screening website). In particular, the information needs recorded by women were frequently interlinked with their personal information and medical history, reflecting attempts to make sense of their results. There are likely to be women who want to discuss their specific results and this might also include how their risk relates to their screening history or other health conditions. For these women, knowing who they can contact (eg, a specific helpline, a cancer charity helpline, their GP) will be important.

Finally, for some women there was confusion about why changes were being made to the screening programme and concern about the fact that their sample had not been checked for abnormal cells. It is important to explain why screening is changing and to reassure women that HPV testing is better than cytology, with the changes being made to improve the screening programme. It might also be useful to clarify specifically that this is safe even for women who are at the end of screening (mentioned by some women in their 60s) or those who have previously had abnormal cytology results. The recent public backlash following changes to the cervical screening programme in Australia has highlighted the importance of explaining the rationale behind and safety of changes being made in public health.[19]

Women taking part in HPV-based cervical screening continue to have additional information needs. Information about the epidemiology of HPV, why the cervical screening procedure is changing, and the meaning and implications of different results should be provided in materials accompanying results. Tailored information and signposting to additional materials and resources would also help to ensure women can find the information they need.

**Acknowledgements** The authors would like to thank all the laboratory staff who made recruiting for the psychological evaluation possible, as well as Kirsty Bennett

and Fatima Osmani who contributed to recruitment and data entry. They also thank Julieta Patnick for her helpful comments.

**Contributors** JW, LM, ASF and HK conceived the study. JW, LM, EM and ASF developed the measures. EM, JW and LR managed the project. All authors contributed to the analyses. LM drafted the paper. All authors contributed to the final version of the manuscript.

**Funding** The study was funded by Public Health England (PHE). No grant number was given. PHE funded Emily McBride from 03 January 2016 until 30 September 2017. EM was funded by the National Institute for Health Research (NIHR) from 01 October 2017 (grant number DRF-2017-10-105); JW, LM and AF are funded by Cancer Research UK (grant numbers C7492/A17219 and C49896/A17429).

**Disclaimer** The views expressed in this article are not necessarily those of the NHS, the NIHR or the Department of Health and Social Care

**Competing interests** None declared.

**Patient consent for publication** Not required.

**Ethics approval** Health Research Authority (HRA) approval was granted on 26 September 2016 (Research Ethics Committee reference: 16/LO/0902 and Confidentiality Advisory Group reference: 16/CAG/0047).

**Provenance and peer review** Not commissioned; externally peer reviewed.

**Data availability statement** Data are available upon reasonable request to the corresponding author.

**ORCID iDs**
Laura Marlow http://orcid.org/0000-0003-1709-2397
Alice S Forster http://orcid.org/0000-0002-9933-7919
Jo Waller http://orcid.org/0000-0003-4025-9132

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
