## [Reviewer comments · BMJ Open]

ARTICLE DETAILS

TITLE (PROVISIONAL)	Information needs among women taking part in primary HPV screening in England: A content analysis
AUTHORS	Marlow, Laura; Forster, Alice; McBride, Emily; Rockcliffe, Lauren; Kitchener, Henry; Waller, Jo

VERSION 1 – REVIEW

REVIEWER	Sveinung Sørbye University Hospital of North Norway, Tromsø, Norway
REVIEW RETURNED	19-Sep-2020

GENERAL COMMENTS	Marlow et al. have explored the information needs among women undergoing HPV primary screening. Women were sent a postal questionnaire shortly after receiving their HPV results and 6 and 12 months later. Each questionnaire asked if women had any unanswered questions about cervical screening or HPV testing. 381 women who recorded one or more free-text responses. The most common theme represented women's emotional responses and attempts to understand their results. General questions about the cause and epidemiology of HPV were raised by 38% of women. Questions about the purpose and procedure for HPV testing were most common among HPV negative women. In conclusion, despite provision of information alongside screening invitations, women can still have unanswered questions following receipt of their results. Details about the epidemiology of HPV and why cervical screening procedures are changing should be included with screening invitations. Some results groups may benefit from additional tailored information with their result letter. The claims are properly placed in the context of the previous literature. The experimental data support the claims. The manuscript is written clearly enough that most of it is understandable to non-specialists. The authors have provided adequate proof for their claims, without overselling them. The authors have treated the previous literature fairly. The paper offers enough details of methodology so that the experiments could be reproduced. Comments: Marlow et al. should include a figure with the UK national guidelines (flow chart) of HPV test in primary screening. Page 4, line 19-21, "These figures are expected to decrease dramatically as cohorts offered HPV vaccination move into the programme" => "These figures are expected to decrease
---

	somewhat as cohorts offered HPV vaccination move into the programme" The bivalent HPV-vaccine (Cervarix) covers HPV type 16 and 18. The HPV DNA screening test Cobas 4800 covers 14 HPV-types. HPV type 16 and 18 accounts for 25 % of all HPV positive results using Cobas 4800. Hence, the HPV test positivity rate are expected to decrease about 25 % when cohorts offered HPV vaccination move into the programme (Brotherton 2019). Page 8, line 28-30, "Women with abnormal cytology seemed to have fewer questions about HPV" This is expected when HPV+/cyt+ women are referred to colposcopy and can talk with a gynecologist. Women with normal cytology (HPV+/cyt-) should wait for retesting after 12 months. Discussion, add, "In a Norwegian study they compared anxiety and depression scores among participants by screening arm. In total, 1,008 women answered a structured questionnaire. The results suggest that a change to hrHPV testing in primary screening would not increase psychological distress among participants." (Andreassen 2019). References Brotherton JM, Hawkes D, Sultana F, Malloy MJ, Machalek DA, Smith MA, Garland SM, Saville M. Age-specific HPV prevalence among 116,052 women in Australia's renewed cervical screening program: A new tool for monitoring vaccine impact. Vaccine. 2019 Jan 14;37(3):412-416. https://pubmed.ncbi.nlm.nih.gov/30551987/ Andreassen T, Hansen BT, Engesaeter B, Hashim D, Støer NC, Tropé A, Moen K, Ursin G, Weiderpass E. Psychological effect of cervical cancer screening when changing primary screening method from cytology to high-risk human papilloma virus testing. Int J Cancer. 2019 Jul 1;145(1):29-39. https://pubmed.ncbi.nlm.nih.gov/30549273/
--	---

REVIEWER	Marie-Hélène Mayrand Université de Montréal et CRCHUM, Canada
REVIEW RETURNED	02-Oct-2020

GENERAL COMMENTS	As HPV testing is replacing Pap testing for cervical cancer screening across settings, having a better understanding of women's information needs is essential. The setting for this study is ideal, as it focuses women attending cervical cancer screening in the early phases of HPV testing implementation, making it possible to include a large group of women with different test results. More details (in text or supplementary file) as to how screening is carried out in England would help the reader establish context: Do women get an invitation letter? If so, what type of information is in that letter? Who performs the cervical sampling? What type of information is provided at that time point? Were there any public health information campaigns to explain the change to HPV testing? Can links to result letters be provided?
--

	The choice of 3 time points to assess information needs is a strength and provides a fuller picture of information needs. Methods are succinctly but clearly described. They are appropriate for the research question. The themes described in the results section are informative and could readily be used to improve the information provided to women. Although useful, the authors are correct in pointing out that it is difficult to know if the identified information needs apply to women who did not leave comments to be analyzed. This is an inherent limit of the design. Future studies, for example with a focus on harder to reach groups, would complement these findings.
--	--

VERSION 1 – AUTHOR RESPONSE

Reviewer: 1

Reviewer Name: Sveinung Sørbye

Institution and Country: University Hospital of North Norway, Tromsø, Norway

Please state any competing interests or state 'None declared': None declared

Marlow et al. have explored the information needs among women undergoing HPV primary screening. Women were sent a postal questionnaire shortly after receiving their HPV results and 6 and 12 months later. Each questionnaire asked if women had any unanswered questions about cervical screening or HPV testing. 381 women who recorded one or more free-text responses. The most common theme represented women's emotional responses and attempts to understand their results. General questions about the cause and epidemiology of HPV were raised by 38% of women. Questions about the purpose and procedure for HPV testing were most common among HPV negative women. In conclusion, despite provision of information alongside screening invitations, women can still have unanswered questions following receipt of their results. Details about the epidemiology of HPV and why cervical screening procedures are changing should be included with screening invitations. Some results groups may benefit from additional tailored information with their result letter.

The claims are properly placed in the context of the previous literature. The experimental data support the claims. The manuscript is written clearly enough that most of it is understandable to non-specialists. The authors have provided adequate proof for their claims, without overselling them. The authors have treated the previous literature fairly. The paper offers enough details of methodology so that the experiments could be reproduced.

Response: Thank you for this positive assessment of our paper

Comments:

Marlow et al. should include a figure with the UK national guidelines (flow chart) of HPV test in primary screening.

Response: Thank you for this suggestion. We have added the flow chart to the supplementary material as Supplementary Figure 1. We have also added a line under Participants as follows: 'Women testing HPV negative were invited for routine recall, whereas those testing positive had reflexive cytology and were managed accordingly (see Supplementary Figure 1).'

Page 4, line 19-21, "These figures are expected to decrease dramatically as cohorts offered HPV vaccination move into the programme" => "These figures are expected to decrease somewhat as cohorts offered HPV vaccination move into the programme."

The bivalent HPV-vaccine (Cervarix) covers HPV type 16 and 18. The HPV DNA screening test Cobas 4800 covers 14 HPV-types. HPV type 16 and 18 accounts for 25 % of all HPV positive results using Cobas 4800. Hence, the HPV test positivity rate are expected to decrease about 25 % when cohorts offered HPV vaccination move into the programme (Brotherton 2019).

Response: We have made the suggested edit from 'dramatically' to 'somewhat'.

Page 8, line 28-30, "Women with abnormal cytology seemed to have fewer questions about HPV"

This is expected when HPV+/cyt+ women are referred to colposcopy and can talk with a gynecologist. Women with normal cytology (HPV+/cyt-) should wait for retesting after 12 months.

Response: We absolutely agree and have added note to this effect in the Discussion, although it is actually unlikely that most women in this group would already have attended for colposcopy at the point where they were completing the questionnaire. We have added the following sentence:

'This may in part be because they had been referred for colposcopy and even if they had not yet attended and had the opportunity to ask questions, they would have received an additional information leaflet with their results letter.'

Discussion, add, "In a Norwegian study they compared anxiety and depression scores among participants by screening arm. In total, 1,008 women answered a structured questionnaire. The results suggest that a change to hrHPV testing in primary screening would not increase psychological distress among participants." (Andreassen 2019).

Though not directly comparable to our findings, we agree that highlighting quantitative studies showing hrHPV testing does not raise clinically significant dispan style="font-family:'Segoe UI'; color:#201f1e">stress, may be helpful for the reader to help gain some context. We have added the sentences:

'Studies exploring the psychological impact of testing HPV positive in the context of organised screening show no differences in distress across result groups, but anxiety can be slightly higher in HPV positive women'.

References

Brotherton JM, Hawkes D, Sultana F, Malloy MJ, Machalek DA, Smith MA, Garland SM, Saville M. Age-specific HPV prevalence among 116,052 women in Australia's renewed cervical screening program: A new tool for monitoring vaccine impact. *Vaccine*. 2019 Jan 14;37(3):412-416.

Andreassen T, Hansen BT, Engesaeter B, Hashim D, Støer NC, Tropé A, Moen K, Ursin G, Weiderpass E. Psychological effect of cervical cancer screening when changing primary screening method from cytology to high-risk human papilloma virus testing. *Int J Cancer*. 2019 Jul 1;145(1):29-39.

Response: We have added the suggested references.

Reviewer: 2

Reviewer Name: Marie-Hélène Mayrand

Institution and Country: Université de Montréal et CRCHUM, Canada

Please state any competing interests or state 'None declared': None declared

As HPV testing is replacing Pap testing for cervical cancer screening across settings, having a better understanding of women's information needs is essential. The setting for this study is ideal, as it focuses women attending cervical cancer screening in the early phases of HPV testing implementation, making it possible to include a large group of women with different test results.

Response: Thank you

More details (in text or supplementary file) as to how screening is carried out in England would help the reader establish context: Do women get an invitation letter? If so, what type of information is in that letter? Who performs the cervical sampling? What type of information is provided at that time

point? Were there any public health information campaigns to explain the change to HPV testing? Can links to result letters be provided?

Response: Thank you for this suggestion. We have added further information alongside the flow diagram requested by Review 1 in Supplementary File 1. We are not able to provide links to results letter text but have included a link to the leaflet women receive with their screening invitation, as well as the information provided to women in the current study. The information provided at the point of screening is hard to determine as this is largely down to the sample-taker.

The choice of 3 time points to assess information needs is a strength and provides a fuller picture of information needs.

Response: Thank you for recognising this strength

Methods are succinctly but clearly described. They are appropriate for the research question.

The themes described in the results section are informative and could readily be used to improve the information provided to women.

Although useful, the authors are correct in pointing out that it is difficult to know if the identified information needs apply to women who did not leave comments to be analyzed. This is an inherent limit of the design. Future studies, for example with a focus on harder to reach groups, would complement these findings.

Response: We agree and have added a comment in the Discussion that future research should focus on harder to reach groups: 'Future research on HPV information needs should focus on these harder to reach groups.'

VERSION 2 – REVIEW

REVIEWER	Sveinung Wergeland Sørbye University Hospital of North Norway Department of Clinical pathology Norway
REVIEW RETURNED	10-Nov-2020

GENERAL COMMENTS	All comments have been addressed. The response to the comments is adequate and improved the manuscript.
---

REVIEWER	Marie-Hélène Mayrand Université de Montréal et CRCHUM Canada
REVIEW RETURNED	24-Nov-2020

GENERAL COMMENTS	Thank you for considering comments/suggestions and adding the pertinent information
---